# Boundless Byte Pair Encoding:
# Breaking the Pre-tokenization Barrier

**Craig W. Schmidt, Varshini Reddy & Chris Tanner**[*]
Kensho Technologies
Cambridge, MA 02138, USA
{craig.schmidt,varshini.bogolu,chris.tanner}@kensho.com

**Yuval Pinter**
Department of Computer Science
Ben-Gurion University of the Negev
Beer Sheva, Israel
uvp@cs.bgu.ac.il

## Abstract

Pre-tokenization, the initial step in many modern tokenization pipelines, segments text into smaller units called *pretokens*, typically splitting on whitespace and punctuation. While this process encourages having full, individual words as tokens, it introduces a fundamental limitation in most tokenization algorithms such as Byte Pair Encoding (BPE). Specifically, pretokenization causes the distribution of tokens in a corpus to heavily skew towards common, full-length words. This skewed distribution limits the benefits of expanding to larger vocabularies, since the additional tokens appear with progressively lower counts. To overcome this barrier, we propose BOUNDLESSBPE, a modified BPE algorithm that relaxes the pretoken boundary constraint. Our approach selectively merges two complete pretokens into a larger unit we term a *superword*. Superwords are not necessarily semantically cohesive. For example, the pretokens ␣of and ␣the might be combined to form the superword ␣of␣the. This merging strategy results in a substantially more uniform distribution of tokens across a corpus than standard BPE, and compresses text more effectively, with an approximate 20% increase in bytes per token.

## 1 Introduction

Pre-tokenization is a crucial step in preparing text for language models, helping to align token boundaries to meaningful linguistic units. A document is first broken into chunks called *pretokens* using a regular expression,[1] which are then tokenized separately. Each pretoken may be tokenized into two or more subword tokens, or used as a single token that exactly matches the entire pretoken. Many common words end up as a single token (Reddy et al., 2025), more than 90% according to our analysis (Section 2). Under such circumstances, differences in the tokenizer itself can only manifest in the small percentage of remaining pretokens. This high overlap in resulting tokens shared across different tokenizers can explain why tokenizers have been found to perform quite similarly on downstream tasks, with no statistically superior approach (Schmidt et al., 2024). Selecting a larger vocabulary size tends to exacerbate this issue, since the lower-frequency tokens added to the vocabulary later in the training process do not substantially change the resulting token distribution in the training corpus.

---

[*]Chris Tanner is also affiliated with MIT in Cambridge, MA, USA.
[1]See Section 3.5 and Appendix A for a discussion of the particular pre-tokenizer regular expression we used.

Zouhar et al. (2023) suggest that tokenizers exhibiting a more uniform distribution of token frequencies across a corpus tend to yield superior performance in language models. They suggest avoiding tokens which exhibit high frequency while carrying little semantic content, such as individual bytes, as well as very rare tokens that lack sufficient contextual information for effective learning. However, current pre-tokenization methodologies offer limited control over this distribution, as most pretokens are mapped to single tokens, often encompassing common whole words.

With the goal of obtaining a more uniform distribution of tokens, we propose a modification to the standard pre-tokenization approach that still allows us to retain the benefits of contextual cohesion. Specifically, we introduce *superwords*, tokens composed of an *n*-gram of words, to supplement subwords and words in the vocabulary. The notion of superwords extends beyond semantically cohesive units like `New York City` often identified through metrics like Pointwise Mutual Information (PMI; Fano & Wintringham, 1961; Church & Hanks, 1989). Instead, they serve to distribute the most common words like ␣the into a range of common *n*-grams like ␣the␣car and ␣the␣house, thereby lowering the frequency of the most common tokens in a corpus. As we will demonstrate, a substantial number of such common *n*-grams is found by our approach, enabling more effective utilization of larger vocabularies.

In Section 3, we introduce BOUNDLESSBPE, an extension to the widely adopted Byte Pair Encoding (BPE) algorithm (Sennrich et al., 2016; Gage, 1994). The key modification involves incorporating the concept of a *supermerge*. During the tokenizer training process, we permit the merging of two adjacent pretokens if each is represented by a single token, in which case, the resulting merge is considered a *superword*. The supermerge operation lends itself to seamless integration into the standard BPE training procedure: at each step of training, the merge or permissible supermerge that occurs the most frequently is performed. We also implement a variation of the deletion technique from PickyBPE (Chizhov et al., 2024) to remove low-frequency intermediate tokens from the vocabulary. These deletions contribute to a more uniform distribution at the lower end of the frequency spectrum by freeing up space taken by intermediate tokens for use by more common tokens.

The efficacy of our proposed approach is presented in Section 4, where we use an evaluation corpus to compare the token frequency distributions produced by BOUNDLESSBPE and several commonly-used baseline tokenizers. The results demonstrate that BOUNDLESSBPE yields a more uniform token distribution, consequently achieving a minimum of a 21% increase in Rényi efficiency (Zouhar et al., 2023) compared to the baselines. Since our method exhibits improved compression, with at least a 19.7% increase in overall bytes per token, it requires fewer tokens for language model inference.

## 2 Limitations of pre-tokenizers

Pre-tokenization is a processing step shared by most tokenizers and it plays a vital role in how tokenizer vocabularies are formed. Regular expressions commonly used in pre-tokenization permit spaces only as the first byte in a pretoken (for example, ␣the), which supports the alignment of the final tokens with word boundaries. While the objective is to generate tokens that correspond closely to meaningful linguistic units, this alignment does not mandate a strict one-to-one mapping between tokens and complete words. Instead, it prioritizes the prevention of tokens containing parts of two adjacent words, thereby enabling better downstream performance. For instance, Schmidt et al. (2024) demonstrated that entirely omitting pre-tokenization resulted in the poorest downstream performance among 18 evaluated tokenizers.

Due to the Zipfian nature of text, where a few words occur with very high frequency and many words occur rarely, the pretokens of common words are often efficiently represented as a single token that exactly matches the entire pretoken. We used a 5GB portion of MiniPILE (Kaddour, 2023) as our out-of-sample evaluation corpus to illustrate this.[2] Using

---

[2]We train tokenizers on the first 170,721 documents of MiniPile, totalling 1GB. The remaining 829,279 documents form the 5GB out-of-sample evaluation corpus used throughout this paper.

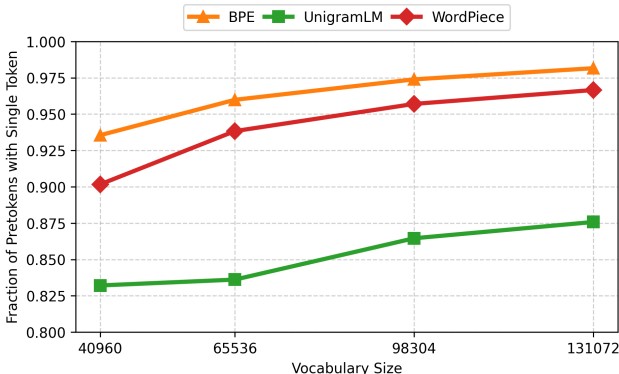

Figure 1: Proportion of pretokens in the evaluation corpus that are represented as full tokens by different tokenization methods (BPE, UnigramLM, WordPiece) across varying vocabulary sizes.

the widely adopted BPE, WordPiece (Wu et al., 2016; Schuster & Nakajima, 2012) and UnigramLM (Kudo, 2018) tokenizers,[3] with vocabulary sizes of 40,960, 65,536, 98,304, and 131,072, we calculated the proportion of pretokens that were ultimately represented by a single token. As shown in Figure 1, BPE and WordPiece, both of which are bottom-up merge-based tokenizers, achieve a single-token representation for well over 90% of pretokens, while UnigramLM, a top-down ablation-based tokenizer, achieves 82-87%. This highlights how these tokenization methods effectively handle the most frequent words in the vocabulary, and it is a direct consequence of the Zipfian distribution inherent in natural language.

The pre-tokenization step is thus directly determining the vast majority of the tokens, leaving the tokenization training very little ability to modify the token distribution. The phrase `'To be or not to be'` composed of very common words will be tokenized by our baseline tokenizers as

`['To', ' be', ' or', ' not', ' to', ' be'],`

simply due to pre-tokenization. To change the tokenization substantially, we need a way to overcome the fact that a pretoken's role during tokenizer training effectively ends once it becomes a single token. For common words, this happens very early in the process.

## 3 BOUNDLESSBPE

We introduce BOUNDLESSBPE, a tokenization algorithm that allows adjacent pretokens, each represented as a single token, to be merged into a superword token. For the Shakespearean example above, our tokenization training process will continue beyond pretoken boundaries, yielding the tokens:[4]

`['To be', ' or not', ' to be'].`

### 3.1 Tokenizer training

Standard BPE training starts with every pretoken in the training data split into individual bytes, which are the initial tokens. For example, `'Tip of the hat'` has four pretokens, each containing single-byte initial tokens:

`[['T', 'i', 'p'], [' ', 'o', 'f'], [' ', 't', 'h', 'e'], [' ', 'h', 'a', 't']].`

BPE employs pairwise merge rules, such as `(' ','t') -> ' t'`, where a token pair is merged to form a single token. At each iteration, the algorithm finds the pair of adjacent

---

[3]We use the Hugging Face implementations: https://github.com/huggingface/tokenizers.
[4]BOUNDLESSBPE examples throughout this paper are with a vocabulary of 131,072.

tokens with the highest frequency of occurrence across the training data. This merge rule with the maximum adjacency count is added to a list of merge rules (used during inference), and the newly-combined token is added to the vocabulary. This pair of tokens is replaced by the combined token throughout the training data, giving:

`[['T', 'i', 'p'], [' ', 'o', 'f'], [' t', 'h', 'e'], [' ', 'h', 'a', 't']].`

This process continues until the vocabulary reaches the desired size. During inference, applicable merge rules are applied to a document in the same order they were found in training, until no further merges are possible, giving the final tokenization.

BOUNDLESSBPE is an extension of this standard BPE training process, with two key modifications. First, two adjacent pretokens that are currently tokenized with a single token (identical to the pretoken itself) are allowed to merge into a combined *superword*. We call these merges *supermerges*. For example, the adjacent pretokens `[' of']` and `[' the']` each consisting of a single token:

`[['T', 'i', 'p'], [' of'], [' the'], [' h', 'at']],`

can be combined into the superword `[' of the']`, which would consist of a single token. A superword can subsequently be merged with other fully-merged pretokens into longer superwords.[5] Supermerges often involve pairs of common words, such as `' of the'`, `' in the'`, `' to the'`, `' on the'`, and `' and the'`. One consequence of this behavior is a reduction in the count of the most common tokens such as `' the'` and `' of'`, as some of their occurrences are allocated to various superwords.

Our second modification to the standard BPE algorithm is the inclusion of a variant of PickyBPE deletions (Chizhov et al., 2024) to eliminate intermediate tokens that primarily serve as components of more frequent, larger tokens. After a regular merge operation, we use PickyBPE's Intersection over Self (IoS) metric to assess the utility of deletion: we compute the ratio of the frequency with which the merged tokens appear consecutively to the individual frequency of each constituent token. If the computed IoS value exceeds a threshold, $\tau = 0.9$, the constituent token is removed from the vocabulary. This deletion step is effective in eliminating low-count tokens because a token with a high IoS predominantly occurs as part of the newly formed token, thereby freeing up space within the desired vocabulary size for higher-frequency tokens. For example, after merging `' bet'` with `'ween'`, the token `'ween'` will be deleted, since its IoS is $0.9618 > 0.9$.

For the sake of simplicity, we do not implement deletions following supermerges, as it would require additional bookkeeping of the initial states.[6] In the original PickyBPE implementation, a deleted token is split back into the pair that formed it. We adopt a more aggressive approach where a deleted token is decomposed back into single bytes within a pretoken. This allows single bytes to potentially recombine into higher-frequency tokens, at the expense of more merge rules.

At any given iteration in the tokenizer training process, we therefore have three potential operations: a regular merge, a supermerge, or a standard deletion. Both regular merges and supermerges with highest frequency are identified, and the operation with the higher score is performed. After each regular merge is selected, one or both merged tokens may then be deleted according to their respective IoS values.

---

[5]To avoid combining different pretoken categories, such as digits and words, we restrict the merging process to pretokens matching the byte regex `rb"^[ _'a-zA-Z]*[a-zA-Z][ _'a-zA-Z]*$"`. This regex matches words with optional initial spaces, contractions, and snake case variables. See Section 3.5 and Appendix A for a more detailed explanation of the pre-tokenization regex. Note that numbers are already segmented into groups of 3 in a right-to-left manner, following the suggestion of Singh & Strouse (2024), so we avoid combining these digit groups into longer superwords to preserve their intended format. Furthermore, as discussed in Appendix A.1, we opted not to combine punctuation with words.

[6]In our training run with a vocabulary size of $2^{17} = 131,072$, there would only have been 164 deletions in the 41,038 supermerges, compared to 1,987 deletions for the 89,778 regular merges. Single bytes are never deleted, as they are necessary to avoid the generation of unknown tokens.

## 3.2 Inference procedure

Once a tokenizer has been trained, it can be used to tokenize a new document, a process often referred to as segmentation or inference. A trained BOUNDLESSBPE tokenizer consists of separate dictionaries for regular merge rules, supermerge rules, and regular deletion rules. Each rule has a unique index representing the order in which it was added during the training process. Each step of inference involves identifying possible operations that can be performed given the current tokenization of the document. Each pair of adjacent tokens within each pretoken is compared to the merge rules. Each pair of adjacent pretokens containing single tokens is compared to the supermerge rules. Finally, each individual token is compared to the deletion rules. The operation with the smallest index, meaning it was the first to be added during the training process, is performed everywhere it occurs within the document. Listing 1 gives an example of this tokenization process for the phrase `'Tip of the hat'`:

```
1  [['T', 'i', 'p'], [' ', 'o', 'f'], [' ', 't', 'h', 'e'], [' ', 'h', 'a', 't']]
2  [['T', 'i', 'p'], [' ', 'o', 'f'], [' t', 'h', 'e'], [' ', 'h', 'a', 't']]
3  [['T', 'i', 'p'], [' ', 'o', 'f'], [' t', 'he'], [' ', 'h', 'a', 't']]
4  [['T', 'i', 'p'], [' ', 'o', 'f'], [' the'], [' ', 'h', 'a', 't']]
5  [['T', 'i', 'p'], [' ', 'o', 'f'], [' the'], [' ', 'h', 'at']]
6  [['T', 'i', 'p'], [' o', 'f'], [' the'], [' ', 'h', 'at']]
7  [['T', 'i', 'p'], [' of'], [' the'], [' ', 'h', 'at']]
8  [['T', 'i', 'p'], [' of'], [' the'], [' h', 'at']]
9  [['T', 'i', 'p'], [' of the'], [' h', 'at']]
10 [['T', ' ip'], [' of the'], [' h', 'at']]
11 [['T', 'ip'], [' of the'], [' hat']]
12 [[' Tip'], [' of the'], [' hat']]
```

Listing 1: Inference example

The data is initialized on line 1 with a list of 4 pretokens, each containing a list of individual bytes as the initial tokens. Line 2 shows the results of the first merge of (`' '`, `'t'`), which had the lowest index of any operation. This process of selecting the lowest index merge, supermerge, or deletion continues until line 7 where [`' of'`] and [`' the'`] appear next to each other, and each is a single token. At this point, a supermerge becomes a valid option. On line 8, a regular merge (`' '`, `'h'`) is performed next due to a lower index, but then the first supermerge creating [`' of the'`] happens on line 9. The tokenization process concludes on line 12 after the application of a few more merges.

## 3.3 Efficient training implementation

In the representation shown in Section 3.2, a document is segmented into pretokens, with each pretoken containing one or more tokens. While conceptually straightforward and directly applicable to tokenizer training, this representation is computationally inefficient. See Appendix B for a description of several techniques used to speed up training time. We will release an open source version of the BOUNDLESSBPE training and inference code.[7]

## 3.4 Training dynamics

Figure 2 shows the logarithm of the count of each selected merge or supermerge during the training process for BOUNDLESSBPE, along with the count of each selected merge for standard BPE and PickyBPE, each employing two distinct pre-tokenization regular expressions. The addition of supermerges as an available choice allows the counts to decrease at a slower rate compared to the baseline methods. As we will demonstrate, this

---

[7]https://github.com/kensho-technologies/boundlessbpe

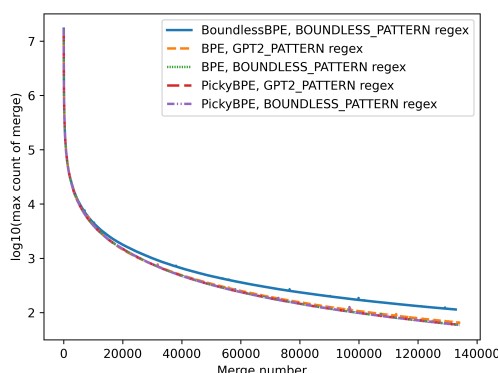

Figure 2: Logarithm of the maximum count of each selected merge, up to a vocabulary size of 131,072.

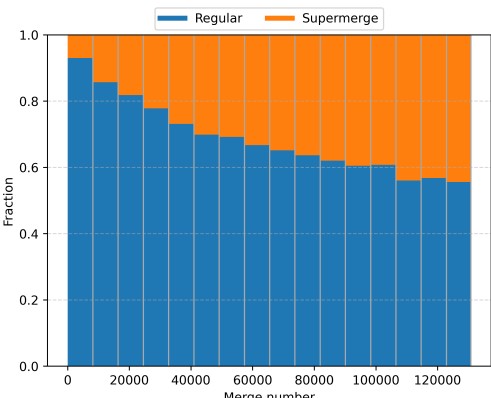

Figure 3: Average fraction of merges (lower) and supermerges (upper) over intervals of 8,192 merges, up to a vocabulary size of 131,072.

results in a more uniform token distribution characterized by a tail of tokens with higher frequencies. The other four curves are essentially indistinguishable, which is a result of so many pretokens ending up as single tokens, as described in Section 2.

Figure 3 shows the fraction of regular merges and supermerges across each successive group of 8,192 merges. A supermerge cannot form until both candidate pretokens are a single tokens. For example, `[' of']` and `[' the']` need to be represented as single tokens `' of'` and `' the'`, respectively, before they are eligible for a supermerge. As a result, early in the process regular merges constitute a majority of operations, and over time the proportion of supermerges grows. Supermerges were 31.3% of total merges over the entire range to a vocabulary size of 131,072. In the rightmost interval, supermerges constitute 44.4% of operations. Thus, the BOUNDLESSBPE tokenizer uses a substantial number of superwords, breaking through the pre-tokenization barrier.

## 3.5 Superwords enable improved pre-tokenization

The `BOUNDLESS_PATTERN` regex in Listing 2 contains a number of improvements over existing pre-tokenization regex patterns.[8] One notable improvement is the better handling of names in code, which is directly enabled by the supermerges in BOUNDLESSBPE. The variable name `'XMLHttpRequest'` is composed of three sub-components: `'XML'`, `'Http'`, and `'Request'`, which can be identified by capitalization conventions. The B-2 to B-6 branches of Listing 2 work together to break variable and function names into smaller pretokens based on capitalization. These sub-components can then be recombined via supermerges as their co-occurrence counts warrant. The example `'XMLHttpRequest snake_case camelCase CONSTANT'` is pre-tokenized as:

`['XML', 'Http', 'Request', ' snake', '_case', ' camel', 'Case', ' CONSTANT'].`

Using the the commonly used `GPT2_PATTERN`[8] pre-tokenization segments on the full names, unable to utilize the prior knowledge encoded in the sub-components:

`['XMLHttpRequest', ' snake', '_', 'case', ' camelCase', ' CONSTANT'].`

Conversely, using `BOUNDLESS_PATTERN` with standard BPE would always result in the final tokens being the individual sub-components. With the supermerges BOUNDLESSBPE provides, it is possible to both align with sub-components and to recombine some of the full names.

---

[8]We discuss existing regular expressions and compare them to ours in Appendix A.

```
BOUNDLESS_PATTERN = "|".join([
  r" ?(?:\p{L}\p{M}*)+['\u2019](?:\p{L}\p{M}*)+",  # B-1, contraction
  r"_(?:\p{Ll}\p{M}*)+",                           # B-2, snake_case
  r" ?(?:\p{Lu}\p{M}*)+(?=(?:\p{Lu}\p{M}*)(?:\p{Ll}\p{M}*))",  # B-3, words
  r" ?(?:\p{Lu}\p{M}*)?(?:\p{Ll}\p{M}*)+",         # B-4, words
  r" ?(?:\p{Lu}\p{M}*)+",                          # B-5, words
  r" ?(?:[\p{Lt}\p{Lm}\p{Lo}]\p{M}*)+",            # B-6, words
  r"(?:\p{N}\p{M}*){1,3}(?=(?:(?:\p{N}\p{M}*){3})*(?:(?:\P{N}\p{M}*)|$))",# B-7
  r"(?:[\p{P}\p{S}]\p{M}*)+",                       # B-8, punct and symbols
  r"[^\S\r\n]*[\n\r]+|[^\S\r\n]+",                  # B-9, whitespace
  r"(?:[\p{Z}\p{C}]\p{M}*)+",                       # B-10, sep or control
  r"\p{M}+"                                         # B-11, leftover marks
])
```

Listing 2: Pre-tokenization regular expressions (regex)

## 4 Token distribution

We have seen that pre-tokenization influences more than 90% of the tokens in the vocabulary, which makes it difficult for algorithms like BPE to alter the distribution of token occurrences. However, supermerges offer a means to overcome this barrier.

The left column of Figure 4 shows the log counts for each token, sorted in descending order of frequency along the $x$-axis, in our evaluation corpus. We observe that the tail of BOUNDLESSBPE's token frequency chart is substantially higher than that of the three baseline tokenizers[9] for two vocabulary sizes,[10] and it has far fewer tokens that don't appear in the evaluation corpus (plotted at zero). This is due to a combination of higher counts for final merges mentioned above and the deletion of intermediate tokens, which makes room for additional useful tokens. The right column of Figure 4 presents a zoomed-in view of this same measure for the 250 tokens with the largest counts in each vocabulary, showing that supermerges have indeed succeeded in reducing the counts of overly-general tokens.

Figure 5 shows the fraction of the vocabulary that is used at least once on the same evaluation corpus. Over 97.5% of tokens in BOUNDLESSBPE's vocabulary are found in the evaluation corpus, across all four vocabulary sizes, while baseline tokenizers' vocabularies include between 82-95%.

Finally, we quantify the token distributions over the corpus using two metrics: (1) the Rényi efficiency metric (Zouhar et al., 2023), which indicates how uniform a token distribution is; and (2) compression rate of the evaluation corpus. Figure 6 shows that the BOUNDLESS-BPE Rényi efficiency is at least 21% above that of the baselines, using the $\alpha = 2.5$ value recommended by Zouhar et al. (2023). There is later work pointing out limitations of Rényi efficiency in predicting downstream performance (Cognetta et al., 2024b). However, our goal was to design a system with a more uniform distribution of tokens, and it does provides a quantitative measure of the uniformity.

Figure 7 shows that BOUNDLESSBPE provides at least a 19.7% increase in overall bytes per token for the 5GB evaluation corpus compared to baseline tokenizers. Compression rate continues to increase at larger vocabularies, indicating it is able to effectively use the additional vocabulary space. While the effect of compression on downstream performance is unclear (Goldman et al., 2024; Ali et al., 2024; Schmidt et al., 2024; Gallé, 2019), having more bytes per token can speed up language model inference, as fewer tokens are needed to process or generate a given text.

---

[9]The baseline tokenizers used the GPT2_PATTERN described in Appendix A as the pre-tokenization regex, chosen because it is the default for the Huggingface tokenizer, and hence very commonly used.

[10]See Figure 8 in Appendix C for two additional vocabulary sizes.

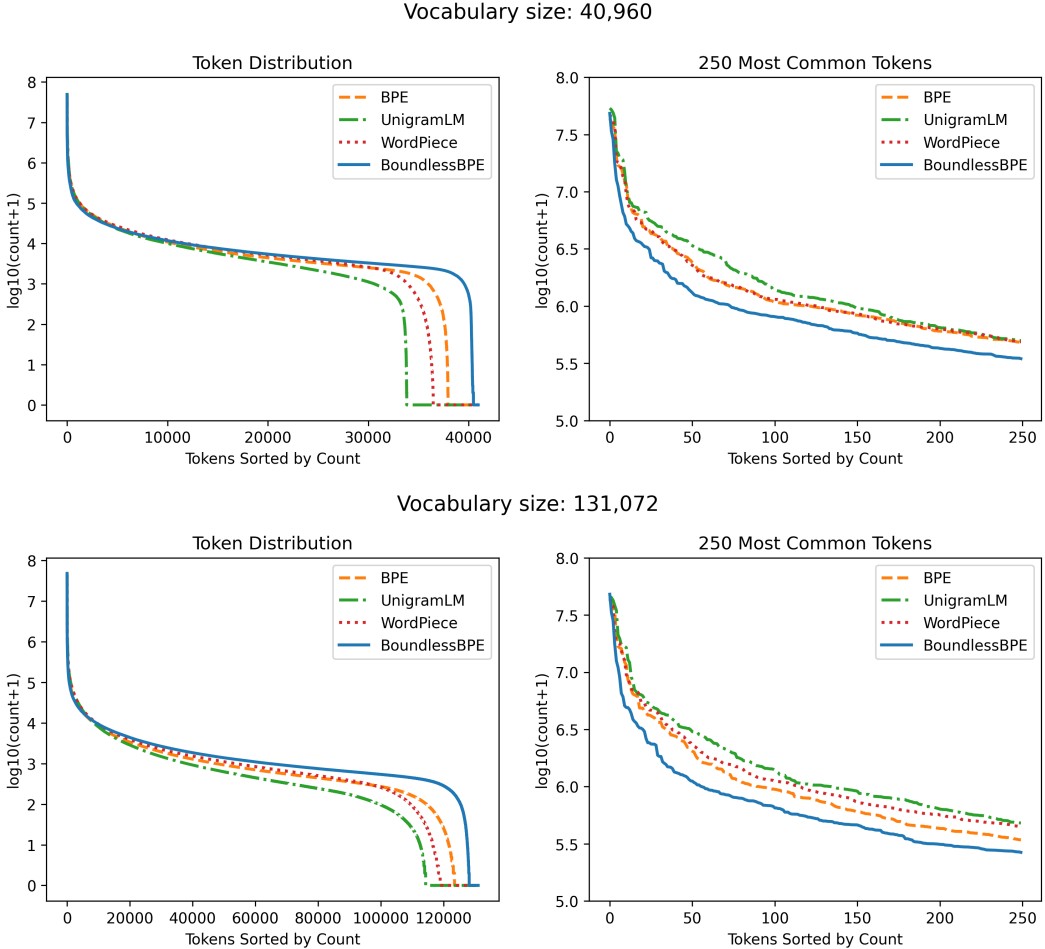

Figure 4: The left column is the $\log_{10}(\text{count} + 1)$ for each token, sorted from most to least frequent on the $x$-axis on our evaluation corpus. The +1 is to allow plotting of 0 counts. The right column shows a zoomed-in view of the 250 most common tokens.

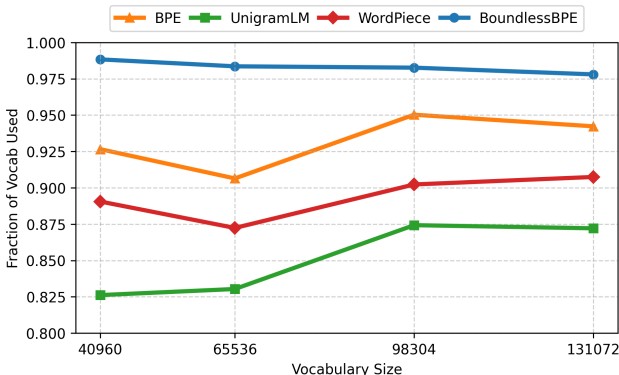

Figure 5: Fraction of vocabulary used at least once in an evaluation corpus, across different tokenization methods and vocabulary sizes. A higher fraction suggests the vocabulary has more useful tokens for representing unseen data.

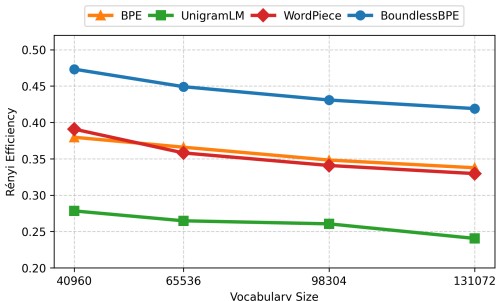 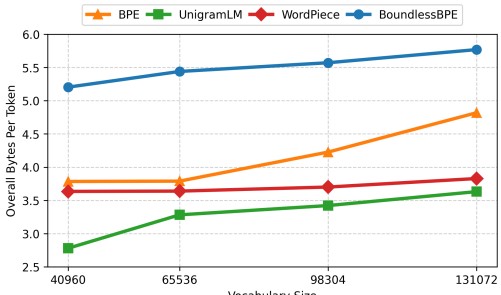

Figure 6: Rényi efficiency, calculated over evaluation corpus, with $\alpha = 2.5$. Tokenizers with higher efficiency are generally desirable.

Figure 7: Bytes per token, calculated over evaluation corpus. A higher value implies better compression, which can lead to efficient training and faster inference.

## 5 Ablation study

BOUNDLESSBPE combined supermerges with a new pre-tokenization regular expression, `BOUNDLESS_PATTERN`. It also used a different form of PickyBPE deletions, where deleted tokens were split into individual bytes before being allowed to re-merge. Our motivation was that the more aggressive decomposition could allow the deleted token to form more compressed tokens. In contrast, Chizhov et al. (2024) break a deleted token into the tokens that created it in a pairwise merge.

| Vocab Size | Regex Pattern | PickyBPE | Bytes Per Token | Rényi Efficiency |
|---|---|---|---|---|
| 40,960 | GPT4o | none | 4.472 | 0.4921 |
| 40,960 | GPT4o | original | 4.497 | 0.4920 |
| 40,960 | GPT4o | ours | 4.492 | 0.4920 |
| 40,960 | Boundless | none | 4.165 | 0.4243 |
| 40,960 | Boundless | original | 4.188 | 0.4242 |
| 40,960 | Boundless | ours | 4.171 | 0.4242 |
| 131,072 | GPT4o | none | 5.072 | 0.4300 |
| 131,072 | GPT4o | original | 5.071 | 0.4300 |
| 131,072 | GPT4o | ours | 5.075 | 0.4302 |
| 131,072 | Boundless | none | 4.635 | 0.3684 |
| 131,072 | Boundless | original | 4.648 | 0.3682 |
| 131,072 | Boundless | ours | 4.645 | 0.3682 |

Table 1: Effect of regular expression pattern and PickyBPE style on intrinsic measures

Table 1 gives an ablation study that disentangles these changes.[11] We consider both forms of PickyBPE deletions, as well as no deletions. We compare our `BOUNDLESS_PATTERN` to a modern baseline of the GPT4o regular expression.[12] We see that the differences in the types of deletions had only a very small effect on either of our intrinsic measures. Deletions may still be important for downstream performance, since low frequency tokens are harder to learn (Su et al., 2024; Yu et al., 2022). The GPT4o regex increased bytes per token by 0.307-0.438. Like `F-2` of `GPT4_PATTERN` discussed in Appendix A, GPT4o allows a single

---

[11]See Appendix D for the same results in two additional vocabulary sizes. The ablations were performed with a slightly different merge eligibility regex `r"^(?=.+pL)(?:pLpM*|[ _'u2019])+$"` that will perform better with Unicode text.

[12]https://github.com/openai/tiktoken/blob/4560a889/tiktoken_ext/openai_public.py#L101-L114

non-alphanumeric character to combine with word tokens, compressing this case more than BOUNDLESS_PATTERN, which was designed to keep character classes more separate.

## 6 Related work

**Impact of pre-tokenization** Velayuthan & Sarveswaran (2025) emphasize the importance of pre-tokenization relative to the tokenizer in achieving egalitarian tokenization across languages. They observe that pre-tokenization limits achievable compression, since each pretoken must contain at least one token. Thus, the number of of pretokens is a lower bound on the number of tokens. Dagan et al. (2024) also showed the substantial impact of pre-tokenization regex choices on tokenizer compression and downstream performance. Furthermore, Wegmann et al. (2025) demonstrate that pre-tokenization has a stronger impact on downstream task performance than that of vocabulary size and training corpus variations.

**Multi-word tokens** Prior work has explored incorporating larger linguistic units into vocabularies. Salehi et al. (2015); Liu et al. (2025); Huang et al. (2025) highlight the benefits of multi-word tokens for compression, training cost, and model performance. Similarly, Otani et al. (2020) show representation improvements using multi-word expressions (MWE's) in multilingual settings. Kumar & Thawani (2022) found that adding high-PMI MWE's improved performance of machine translation better than high-frequency subword or whole-word tokens. Gee et al. (2023) introduced a multi-word tokenizer that augments a standard BPE vocabulary with MWE's by representing frequent $n$-grams as a single token.

In concurrent work, Liu et al. (2025) proposed SuperBPE, an enhancement to BPE that employs a two-pass tokenization strategy to obtain multi-word tokens, which they also term *superwords*. Their method involves an initial BPE training phase with pre-tokenization, conducted up to a vocabulary size $t < T$, where $T$ represents a hyperparameter they call the *transition point*. This phase is followed by a second BPE training pass that resumes from the first but omits pre-tokenization, thereby enabling the formation of superwords to populate the remainder of the vocabulary. In contrast to SuperBPE, BOUNDLESSBPE operates in a single pass rather than in separate stages, allowing both standard merges and supermerges to occur at the same points based on their respective frequencies. Our approach thus obviates the need for a transition point, avoiding a hyperparameter search and speeding up tokenizer training. Additionally, our method offers control over which pretokens can be merged together, preventing different types of pretokens, such as words and punctuation, from merging.

See Appendix E for additional related work.

## 7 Conclusion

While natural language processing has achieved significant advancements in performance over the past decade, certain design choices remain static, such as tokenization algorithms and pre-tokenization regular expressions. Pre-tokenization exerts considerable influence over a corpus token distribution, with standard pre-tokenization methods fixing over 90% of words to be represented as single tokens. To address the current limitations in achieving a uniform token distribution over a corpus, we introduce two key contributions. First, we present BOUNDLESSBPE, a modified BPE training process that enables the combination of adjacent full pre-tokens into superwords. The incorporation of superwords yields enhanced compression, quantified by an increase in bytes per token, and a more uniform distribution of token frequencies across a corpus. Second, we propose BOUNDLESS_PATTERN, a novel pre-tokenization regular expression designed to work with superwords, resulting in improved tokenization, particularly for code and named entities, when compared to standard regular expressions such as the one used by GPT-2. Based on prior work (Zouhar et al., 2023; Liu et al., 2025) we hypothesize that the intrinsic performance improvements shown by BOUNDLESSBPE will have a positive impact on the downstream performance of language models.

## Acknowledgments

Many thanks to Ilya Yudkovich and Mike Arov at Kensho Technologies for their technical assistance. Thanks also to Seth Ebner, Charles Lovering, and Michael Krumdick at Kensho Technologies for many helpful comments and discussions. This research was supported in part by the Israel Science Foundation (grant No. 1166/23).

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

## A Pre-tokenizer regular expression

Pre-tokenization applies a regular expression (regex) to each document to form pretokens, which are then each tokenized separately. The regex in this section make extensive use of Unicode categories. These are not supported by the re library used by default in python, instead requiring the more powerful regex library.[13] In the notation of regex, Unicode can be divided into the categories given in Table 2.

| Category | Description |
|---|---|
| \p{L} | Letters |
| \p{N} | Numbers |
| \p{Z} | Whitespace and other separators |
| \p{S} | Symbols |
| \p{P} | Punctuation |
| \p{C} | Control characters |
| \p{M} | Combining marks (diacritical marks, etc.) |

Table 2: Unicode Character Categories

Any regular expression can be used for pre-tokenization, provided that it matches all of the text in a given Unicode string. Thus, at least one branch of the regex must match each of these Unicode categories.

Previous regex were used without much emperical justification. Dagan et al. (2024) and Wegmann et al. (2025) are among the first to more systematically investigate the effect of the regex on downstream performance.

A number of regex are shown in Listing 3. It includes the regex for GPT-2 and GPT-4,[14] the Punct regex (Dagan et al., 2024), and the proposed regex for BOUNDLESSBPE. Each of the regular expression branches shown in the labeled lists are combined together with the | operator into a single regex. In many of the regex, an initial r" ?" indicates an optional initial space, while a final r"[\r\n]*" indicates zero or more line endings.

### A.1 Separation of Unicode Classes

One open question with the pre-tokenization regex is how much the Unicode character classes should be kept separate. The GPT2_PATTERN pattern kept them very separate. The GPT4_PATTERN pattern moved away from that with pattern in line F-2 that allowed any single character besides a letter, digit, or line ending to come before a word. It also combined line endings with other characters in F-4. To study if this was a good idea, the Punct pattern

---

[13]https://pypi.org/project/regex/

[14]Taken from https://github.com/karpathy/minbpe

```python
import regex as re

GPT2_PATTERN = "|".join([
    r"'(?:[sdmt]|ll|ve|re)",        # T-1, English contractions
    r" ?\p{L}+",                     # T-2, words
    r" ?\p{N}+",                     # T-3, digits
    r" ?[^\s\p{L}\p{N}]+",          # T-4, not letters, digits, or whitespace
    r"\s+(?!\S)",                    # T-5, all-but-last whitespace
    r"\s+"                           # T-6, whitespace
])

GPT4_PATTERN = "|".join([
    r"'(?i:[sdmt]|ll|ve|re)",       # F-1, English contractions
    r"[^\r\n\p{L}\p{N}]?+\p{L}+",   # F-2, words, w/ opt non-alphanumeric
    r"\p{N}{1,3}",                   # F-3, digits
    r" ?[^\s\p{L}\p{N}]++[\r\n]*",  # F-4, not letters, digits, or whitespace
    r"\s*[\r\n]",                    # F-5, whitespace with line-ending
    r"\s+(?!\S)",                    # F-6, all-but-last whitespace
    r"\s+"                           # F-7, all whitespace
])

PUNCT_PATTERN = "|".join([
    r" ?\p{L}+",                     # P-1, words
    r"\p{N}{1,3}",                   # P-2, digits
    r" ?[^\s\p{L}\p{N}]+[\r\n]*",   # P-3, not letters, digits, or whitespace
    r"\s*[\r\n]+",                   # P-4, whitespace with line-ending
    r"\s+(?!\S)",                    # P-5, all-but-last whitespace
    r"\s+"                           # P-6, whitespace
])

BOUNDLESS_PATTERN = "|".join([
    # B-1, contraction, allow curly apostrophe
    r" ?(?:\p{L}\p{M}*)+['\u2019](?:\p{L}\p{M}*)+",
    # B-2, snake_case, with underscore at front
    r"_(?:\p{Ll}\p{M}*)+",
    # B-3, Uppercase, followed by uppercase and lowercase letter
    r" ?(?:\p{Lu}\p{M}*)+(?=(?:\p{Lu}\p{M}*)(?:\p{Ll}\p{M}*))",
    # B-4, optional uppercase, and one or more lowercase
    r" ?(?:\p{Lu}\p{M}*)?(?:\p{Ll}\p{M}*)+",
    # B-5, all uppercase acronym CONSTANT
    r" ?(?:\p{Lu}\p{M}*)+",
    # B-6, titlecase, modifier, or uncased letters
    r" ?(?:[\p{Lt}\p{Lm}\p{Lo}]\p{M}*)+",
    # B-7, numbers
    r"(?:\p{N}\p{M}*){1,3}(?=(?:(?:\p{N}\p{M}*){3})*(?:(?:\P{N}\p{M}*)|$))",
    # B-8, punctuation and symbols
    r"(?:[\p{P}\p{S}]\p{M}*)+",
    # B-9, whitespace
    r"[^\S\r\n]*[\n\r]+|[^\S\r\n]+",
    # B-10, separator or control
    r"(?:[\p{Z}\p{C}]\p{M}*)+",
    # B-11, leftover marks, just for bad utf-8
    r"\p{M}+"
])
```

Listing 3: Comparison of pre-tokenization regular expressions (regex)

(Dagan et al., 2024) returned to only allowing a space before a word with P-1. They found an improvement for PUNCT_PATTERN over GPT4_PATTERN at a vocabulary size of 32k, but no significant difference at a vocabulary size of 80k. Wegmann et al. (2025) also had mixed results on this question. They found the GPT2_PATTERN with the highest separation was best for tasks requiring robustness to language variation, while more mixing was beneficial to tasks requiring sensitivity to language variation.

In the face of mixed evidence, we keep our classes well separated, in the hope that this will give more of the word based tokens that can be combined by super merges.

## A.2 Combining Marks and Unicode Normalization

The T-2, F-2 and P-1 word patterns all have a flaw in the handling of combining marks in the \p{M} class. These patterns only match letters, which then end up in a separate pretoken from any combining marks modify that letter. For example, é, which can be represented as `'e\u0301'` becomes two separate pretokens. The BOUNDLESS_PATTERN patterns keep all combining marks with their base character. The pattern `r"(?:\p{L}\p{M}*)"`, for example, is a letter and zero or more combining marks. These are wrapped in a non-capturing group, so the unit can be treated like a single character. Combining marks are possible with all other Unicode classes, so this approach is used all across the BOUNDLESS_PATTERN patterns. The final branch B-11 of `r"\p{M}+"` matches combining marks without a base letter. For example the combining acute accent `'\u0301'` by itself is a valid Unicode code point, but is linguistically ill-formed without a base character.

The other approach to fix this problem is Unicode normalization. The é can also be written as a single pre-composed character `'\u00E9'`. Unicode normalization can be used to convert between these forms. Gorman & Pinter (2025) show that normalization can fix this type of problems with combining marks. However, Dagan et al. (2024, App. D) argue against normalization, as it is usually non-reversible. Our approach of always keeping combining marks with their base characters solves the problem in a reversible way.

Velayuthan & Sarveswaran (2025) present a more general problem with extended grapheme clusters in Tamil, Sinhala, and Hindi being broken up by pre-tokenization. Some (but not all) graphemes are formed by a base character and one or more combining marks, so this would keep some of their graphemes intact.

## A.3 Code Related Pretokens

Patterns B-2 to B-6 collectively provide a new form of token alignment for variable and function names in code. Programmers follow strict case conventions that allow the patterns to find the individual words within the names, in a programming language independent way. The example `'XMLHttpRequest snake_case camelCase CONSTANT'` becomes:

`['XML', 'Http', 'Request', ' snake', '_case', ' camel', 'Case', ' CONSTANT']`.

Note that these are aligned with parts of the names. With regular BPE, breaking variable names up in pre-tokenization would be prevent the full names from becoming tokens. However, with superword merges these more aligned pretokens have the opportunity to recombine and form the complete variable names. Less common variable names will become several tokens. This is one example of how having supermerges allows more extensive pretoken alignment.

## A.4 Whitespace

The existing whitespace regex (T-5 to T-6, F-5 to F-7, and P-4 to P-6) have small variations, but all use negative lookahead to cause the match to backtrack one space.

With the example `'Hello    world    \n\n  \n    '` and GPT4_PATTERN we get:

`['Hello', '   ', ' world', '     \n\n  \n', '    ']`.

The F-6 pattern matches the first three of the four spaces between Hello and world. Then F-2 picks off the third space in ' world'. The F-5 matches multiple lines of whitespace ending in a line-ending, and then finally F-6 picks up the remaining trailing whitespace on the final line.

Breaking the last space off from longer runs of whitespace increases the number of words preceded by a space. However, runs of multiple spaces are often encountered in the context of code. Especially for space sensitive languages like python, splitting four spaces into three and one would seem to increase the difficulty of coding tasks. Similarly, having white space span multiple lines, as in this example, seems to make coding tasks more difficult. So in contrast to the other approaches, B-9 keeps runs of whitespace together, and breaks multiple lines of whitespace into separate tokens with one or more line endings. For the example we would have:

```
['Hello', '    ', 'world', '    \n\n', ' \n', '    '].
```

This is an area that would benefit from further experimentation.

### A.5 Numbers

Singh & Strouse (2024) take a detailed look at the tokenization of numbers. Early models used patterns like T-3, that just used whichever numbers were found by BPE. As they describe, models then switched to runs of 1 to 3 digits (F-3, P-2) or to using single digits. By adding commas to numbers in the input context to enforce right-to-left tokenization they saw a dramatic decrease in arithmetic errors. However, this can be done directly with regex B-7 without the need for inserting commas, so that '1234567' becomes the pretokens ['1', '234', '567'].

### A.6 Contractions

The T-1 and F-1 patterns match English specific contraction endings like 've or 'll. Disliking the English-specific nature, this was omitted from Punct. Pattern B-1 is more language-independent. It keeps the ending of the contraction together with the word as a single pretoken. Thus, any word containing a straight or curly apostrophe internal to the word is matched, like C'est or J'ai.

## B Efficient implementation

In the representation shown in Section 3.2, a document is divided into pretokens, each of which contains one or more tokens. This is simple conceptually and can be used directly for training, but is extremely slow. Tokenizer training routines commonly use the trick of aggregating the pretokens produced over a large training corpus, and use the aggregate counts when calculating pairwise merge counts. Thus the pretoken [' the'] is only ever tokenized once even though it might appear hundreds of thousands of times. This speedup is crucial for performance reasons. [15]

This allows a faster implementation of BOUNDLESSBPE. We keep two separate sets of the training data. We pre-tokenize and tally up the frequency of the pretokens for regular merges. The aggregation here results in more than a 10x speedup on the training time. This first set of data is initialized with single bytes, and merges are selected according to the total aggregated counts.

We have a separate second copy of the training data for the supermerges, where each document is initially broken into pretokens, which will combined into superwords. Since most documents are distinct, no aggregation can help speed up supermerges.

---

[15]We have our own implementation of BPE with deletions using this speedup, based on Andrej Karpathy's minbpe library. https://github.com/karpathy/minbpe

These two representations are related through the process of *unlocking* pretokens. The first subword representation proceeds as in regular BPE. However, when a pretoken there is reduced to a single token, we say that that pretoken has been unlocked.

The second superword representation can only consider a pairwise merge when two adjacent words are unlocked. After a new pretoken is unlocked, the counts of the superword must be updated accordingly. We track the pairwise and single counts of each representation. For speed, we dynamically compute the changes in counts that result from performing each merge. Despite these optimizations, it still took 4.7 CPU days to train our 1GB training dataset due to the lack of aggregation for supermerges. For comparison, training Hugginface BPE on the same data, vocabulary size, and machine takes 59s.

## C  Token distribution at additional vocabulary sizes

Figure 8 shows the same plot as Figure 4 at the vocabulary sizes of 65,536 and 98,304. The trends are largely the same as in Figure 4, with BOUNDLESSBPE having higher counts at the low frequency end of the distribution compared to baselines, and lower frequencies for the most common tokens. Both these are desirable to have a more uniform distribution.

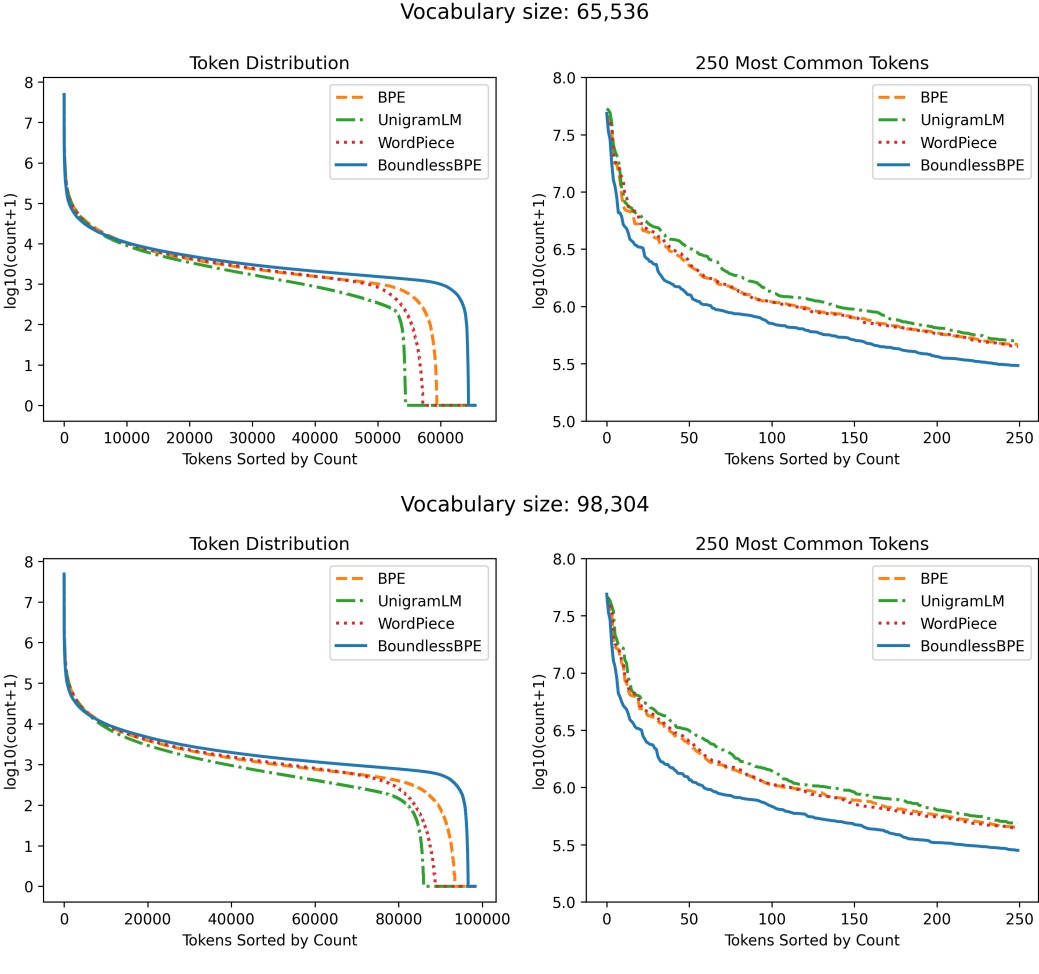

Figure 8: Left column is the $\log_{10}(\text{count} + 1)$ for each token, sorted from most to least frequent on the *x*-axis on our evaluation corpus. The +1 is to allow plotting of 0 counts. Right column shows a zoomed-in view of the 250 most common tokens.

# D  Ablation results at additional vocabulary sizes

| Vocab Size | Regex Pattern | PickyBPE | Bytes Per Token | Rényi Efficiency |
|---|---|---|---|---|
| 65,536 | GPT4o | none | 4.722 | 0.4652 |
| 65,536 | GPT4o | original | 4.722 | 0.4651 |
| 65,536 | GPT4o | ours | 4.735 | 0.4651 |
| 65,536 | Boundless | none | 4.358 | 0.3996 |
| 65,536 | Boundless | original | 4.366 | 0.3995 |
| 65,536 | Boundless | ours | 4.370 | 0.3996 |
| 98,304 | GPT4o | none | 4.927 | 0.4440 |
| 98,304 | GPT4o | original | 4.917 | 0.4438 |
| 98,304 | GPT4o | ours | 4.953 | 0.4438 |
| 98,304 | Boundless | none | 4.549 | 0.3807 |
| 98,304 | Boundless | original | 4.530 | 0.3806 |
| 98,304 | Boundless | ours | 4.542 | 0.3805 |

Table 3: Effect of regular expression pattern and PickyBPE style on intrinsic measures for vocabulary sizes of 65,536 and 98,304

# E  Additional related work

**Foundational Subword Tokenization Algorithms**   Byte level subword tokenization has become a fundamental component in modern NLP, balancing vocabulary size with morphology and out-of-vocabulary handling. Byte Pair Encoding (BPE; Sennrich et al., 2016) iteratively merges the pair of adjacent tokens with the highest count to build a vocabulary. WordPiece (Schuster & Nakajima, 2012) is similar to BPE, except that merges are selected according to their Pointwise Mutual Information (PMI). Kudo (2018) introduced the UnigramLM tokenizer, a top-down approach that starts with a large vocabulary and prunes tokens based on their contribution to sequence likelihood according to a unigram language model.

**Removal of Intermediate Tokens**   The pairwise nature of BPE merges mean some *scaffold* tokens added to the vocabulary simply serve as a bridge to a more popular token, but are not often used on their own. Bostrom & Durrett (2020) observed a dead zone of such tokens in both UnigramLM and BPE vocabularies. The proportion of such tokens were found to be higher in the case of BPE, motivating research into their removal and vocabulary refinement. In Lian et al. (2024) scaffold tokens are marked during a training process, and split into components as a postprocessing step during inference. Chizhov et al. (2024) integrate the deletion step into the BPE training process, where it can affect later merge decisions. Thus an ordered combination of merges and deletions occur during inference. We find Chizhov et al. (2024)'s approach particularly compelling due to its direct integration into BPE training. In contrast, Bauwens & Delobelle (2024) focused on a post-processing step to remove merge rules decreasing morphological alignment. We opted against a purely post-processing approach like Cognetta et al. (2024a) to maintain tighter control over vocabulary construction during training.

