# OpenReview forum: "Boundless Byte Pair Encoding: Breaking the Pre-tokenization Barrier"
_colmweb.org/COLM/2025/Conference — COLM 2025_

### Official Review · Reviewer_s1uu · 2025-04-16

**Rating:** 6
**Confidence:** 3
**Ethics Flag:** 1

**Summary:**

The submission proposes a modification to the standard pre-tokenization approach. The paper introduces superwords, which are tokens of n-gram. The goal is to lower the frequency of the most common tokens in a corpus, enabling more effective utilization of larger vocab. The method is called BOUNDLESSBPE, which performs the most frequent merges during BPE training. Deletion is also considered. Experiments show that the proposed method yield a more uniform token distribution and improved compression.

Quality: The submission is well motivated and executed for the methodology part. The experiments look very brief unfortunately.

Clarity: The reviewer finds the submission written clearly.

Originality: The reviewer is ok with this dimension. The method is simple which may need deep insight.

Significance: The paper may have good potential significance given the importance of tokenizations.

Please see detailed pros and cons below.

**Reasons To Accept:**

1. Pre-tokenization is an important component of language models. The paper provides good motivations on the limitations of existing methods, and the proposed method is simple and sensible.

2. The proposed method is simple, requiring non-trivial insight to the problem.

**Reasons To Reject:**

Overall the reviewer feels the experiments of the submission is lacking.

1. The evaluations are limited to tokenization itself, not any downstream tasks using LM. The experiments are essentially just validated on one small dataset. These limit the significance of the work at its current form.

2. Baselines compared are limited to the classical ones (Wordpiece, BPE, UnigramLM) but not more competitive and recent ones. It's unclear why PickyBPE is discussed, borrowed from, but not directly and fairly compared with.

3. Experiments are limited to EN. This seems to be very limited for a tokenization work.

---

> ### Author Response · Authors · 2025-06-02
>
> Overall, we would like to thank you for your review. You clearly spend the time to understand our work, and we appreciate that. Here are more detailed responses to your comments and questions:
>
> > Baselines compared are limited to the classical ones (Wordpiece, BPE, UnigramLM) but not more competitive and recent ones. It's unclear why PickyBPE is discussed, borrowed from, but not directly and fairly compared with.
>
> Our idea was that the increase in number of merges that occur when doing superwords might cause an increase in junk or unreachable tokens, and the adding PickyBPE would be a good way to counteract that. We do consider PickyBPE by itself in Figure 2 with two different regex, finding very little difference in the number of merges at each step compared to other baselines.  This suggests that the compression ratio will also be quite similar.
>
> > Experiments are limited to EN. This seems to be very limited for a tokenization work.
>
> That is one of the primary limitations of our work.  We should have explicitly stated that as a limitation.

---

> > ### Comment · Reviewer_s1uu · 2025-06-05
> >
> > Thank you for the reply. I think the discussion about PickyBPE helps clarification. The limitations in terms of evaluations and language still persist so I would keep the rating.

---

### Official Review · Reviewer_SuJi · 2025-04-25

**Rating:** 8
**Confidence:** 5
**Ethics Flag:** 1

**Summary:**

This paper highlights the limitation of common pre-tokenization practices, namely splitting on whitespaces and punctuation. The authors propose a modified BPE algorithm to address this issue. While I highlight some concerns about the validity and generalizability of the results, I feel that this proposed approach to tokenization is exciting and a valuable contribution to the literature. As the authors plan to release their code, they are enabling others to study the features and benefits of the proposed tokenization algorithm.

**Questions To Authors:**

* Given the lack of downstream task evaluations, I would have liked to see more analyses of the tokenizer vocabulary. The authors report a small number of examples of different types. For example, how many of the superword tokens start with '_the'? How many of the superword tokens are full phrases (as opposed to parts of phrases missing one or more subword tokens)? This could be tested by looking at the tokenized corpus and seeing how many times a superword token is preceded by or followed by a space, effectively testing whether BoundlessBPE successfully creates full-phrase superword tokens. Other work (https://aclanthology.org/2023.eacl-main.45/, https://arxiv.org/pdf/2409.04599?) analyzes the number or proportion of word-initial tokens. This type of analysis could also make the paper richer.
* Is there a way to evaluate the nature of the multi-word tokens? How much of the time is it cases like '_of_the', versus awkward part-phrases? How many are named entities like 'New York City' versus more functional ones like '_of_the'?
* How much slower is implementation this than Hugging Face’s BPE implementation? The authors report that training took 4.7 CPU days, which seems very long. Is it possible to overcome this so that it can be used for a language model?
* Is this tokenization method sensitive to the quantity of training data? Based on the appendix, the authors trained the tokenizer on 1GB of data. Recent work shows that up to around 150GB there is a benefit to training on more data (https://arxiv.org/pdf/2502.20273; note: the authors cite this paper, but do not discuss that aspect of the paper).

Misc:
* L175 open source → open-source

**Reasons To Accept:**

* The combination of superword tokenization with vocabulary tripping via the IoS metric is extremely interesting. One potential downside of superword merges is that it could create long tokens which are rarely seen. Additionally, as superword tokens will be longer, they will require more merges, increasing the chance of useless intermediate tokens. This is reflected in the lower number of unreachable* tokens.
* This seems like one of the most significant proposed modifications of the BPE algorithm, which is exciting and contributes to a growing line of work seeking more optimal tokenization algorithms.

*I recommend using the term ‘unreachable’ or mentioning its usage in https://arxiv.org/pdf/2405.05417.

**Reasons To Reject:**

* The authors speculate in the conclusion that they believe this tokenization approach would result in improved downstream performance; however the authors do not train a model in order to test this. The authors could try to estimate this using average sentence log probability (proposed in https://arxiv.org/pdf/2109.07306), which was shown to be highly predictive of downstream performance. Currently, it is unclear whether changes to tokenization would make significant performance changes, especially as Schmidt et al. 2024 (https://arxiv.org/pdf/2402.18376) show that in many cases changes to tokenization lead to minimal changes in downstream performance.
* The results are only for English and are only on one dataset (presumably, though this isn’t stated). The results would be more convincing if the authors trained multiple tokenizers on different datasets (or even subsets of the same large dataset) in order to demonstrate that these effects are robust and not specific to any one dataset.
* Some key methodological details remain unclear. What data is the tokenizer trained on? And what is it evaluated on? Is it the same data mentioned in Section 2?
* I think the way that Renyi entropy is used and discussed is potentially problematic, as it does not always correspond with better performance (https://arxiv.org/pdf/2402.14614; https://arxiv.org/pdf/2408.11443). The authors should include more discussion so as not to give the impression that Renyi entropy can be completely trusted as a metric predictive of better performance.

---

> ### Author Response · Authors · 2025-06-02
>
> # SuJi response
>
> Overall, we would like to thank you for your detailed review. You clearly spend the time to understand our work, and we appreciate that. Here are more detailed responses to your comments and questions.
>
> > The combination of superword tokenization with vocabulary tripping via the IoS metric is extremely interesting. One potential downside of superword merges is that it could create long tokens which are rarely seen.
>
> Our training approach will not produce long tokens that are rarely seen, as they are only merged if their count in the training corpus warrants it.  Although intermediate tokens may end up with low counts - which we are using deletions to handle. We will update our terminology on unreachable tokens, and cite that paper.
>
> > Currently, it is unclear whether changes to tokenization would make significant performance changes, especially as Schmidt et al. 2024 show that in many cases changes to tokenization lead to minimal changes in downstream performance.
>
> As we discuss in Section 2, the likely cause of the similarity in performance found by Schmidt et al. 2024 is the huge fraction of pre-tokens tokens that end up as single tokens. This is precisely the motivation for supermerges in BoundlessBPE.  By overcoming this pretokenization barrier, we can substantially change our intrinsic metrics.  The concurrent work of Liu et al (2025) that we cite found sizable downstream improvement across a broad range of tasks.
>
> > The results are only for English and are only on one dataset...
>
> We did only train and evaluate on disjoint parts of MiniPile (see footnote 2, page 3).
>
> > Some key methodological details remain unclear. What data is the tokenizer trained on? And what is it evaluated on? Is it the same data mentioned in Section 2?
>
> We mention that we used the English language dataset MiniPile in footnote 2 on page 3.  We train on the first 1GB of data (due to the slowness of our training code), and evaluate on the remaining 5GB of MiniPile
>
> > I think the way that Renyi entropy is used and discussed is potentially problematic, as it does not always correspond with better performance...
>
> You’re absolutely right that there is mixed evidence on the effect of Renyi efficiency on downstream performance.  We will add the citations you mention.  We were trying to design a system with a more uniform distribution of tokens, and at least it provides a quantitative measure of that, but it may not translate to downstream performance.
>
> > ... For example, how many of the superword tokens start with '_the'? How many of the superword tokens are full phrases (as opposed to parts of phrases missing one or more subword tokens)? ...
>
> In total there are 1388 superwords starting with `_the`.  Here are some examples, with their associated counts
>
> ```text
> _the_same 43131
> _the_first 32755
> ...
> _the_song 132
> ```
>
> All of the superwords are full phrases, due to the fact that we only allow word-like pre-tokens (matching the regex in footnote 5 on page 4) to combine into superwords, if we use that regex as our definition of “word”.
>
> > Is there a way to evaluate the nature of the multi-word tokens? How much of the time is it cases like '_of_the', versus awkward part-phrases?
>
> As previously mentioned, they are all composed of complete pre-tokens, although usually not semantically meaningful phrases.  The most common are listed below:
>
> ```text
> _of_the 975970
> _in_the 606600
> _to_the 394204
> ```
>
> > How many are named entities like 'New York City' versus more functional ones like '_of_the'?
>
> Proper nouns turn out to be fairly infrequent.  A quick analysis of proper nouns within the superwords (regular expressions on the capitalization, and a hand editing pass) found only 475 proper nouns, with a total count on the training corpus of 242,074.  This is out of the overall total of 47,884 unique superwords with a count of 35,506,759.  Thus proper nouns only account for about 1% of the unique superwords, and 0.7% of the counts.  A few examples:
>
> ```text
> _United_States 27932
> _New_York 15985
> _Supreme_Court 7835
> ```
>
> > How much slower is implementation this than Hugging Face’s BPE implementation? The authors report that training took 4.7 CPU days, which seems very long. Is it possible to overcome this so that it can be used for a language model?
>
> Training Hugginface BPE on the same data, vocab, and machine takes 59s. We are currently working on speeding up our code to allow larger training datasets. However, that limitation only applies during training, and so it can be used for LLM use.
>
> > Is this tokenization method sensitive to the quantity of training data? ...
>
> You are correct that we are probably training on a suboptimal amount of training data. We do this to keep our training time - which is linear in the size of data - to a reasonable level.  We will explicitly mention this in the paper.  We are investigating other approaches to speed up superwords training which will allow us to train on substantially more data.

---

> > ### Comment · Reviewer_SuJi · 2025-06-05
> > **Response to Authors**
> >
> > Overall, I am satisfied with the authors’ responses. I will keep my score the same.
> >
> > >We mention that we used the English language dataset MiniPile in footnote 2 on page 3
> >
> > I think this should be stated more clearly in the main text.
> >
> > I don’t know if there is space in the paper, but I think some discussion of the examples you provide of the nature of the superwords would be very interesting (at least in an appendix). Regarding the cases like _of_the and _in_the, these are not semantically meaningless, but rather key grammatical phrases that would be predicted to co-occur by constituency rules. I think this would be of interest to those doing linguistically motivated analyses of tokenizers.
> >
> > >Training Hugginface BPE on the same data, vocab, and machine takes 59s.
> >
> > Could you add this to the part of the paper where you note the training time? I think this will help give readers context.

---

> > > ### Author Response · Authors · 2025-06-09
> > >
> > > We will add the dataset information and BPE timing to the main text, as you suggest.

---

### Official Review · Reviewer_sXwb · 2025-05-11

**Rating:** 4
**Confidence:** 4
**Ethics Flag:** 1

**Summary:**

The authors suggest to extend BPE with "superword" merges and provide some experimental evaluations that hint that the proposed tokenization is more efficient.

**Questions To Authors:**

1. How would the proposed method affect multilingual models?

2. Could you, please, clarify your logic here:

"We first pretrain 8B models with BPE and SuperBPE tokenizers. We use the OLMO2 7B (OLMo et al., 2024) training configuration, including the model architecture, training hyperparameters, and pretraining corpus, but reduce the total number of training steps to correspond to ∼330B tokens (compared to 4T) for the sake of compute."

For reference, out of the recent models Qwen 3 600m parameters has 6T tokenizer. Could you explain your reasoning in detail here? Is there any reason why reduced amount of tokens makes your results more conclusive?

**Reasons To Accept:**

It's a result that would be particularly interesting in the context of "tokenization vs. tokenless" debate. The proposed idea is elegant and the evaluations hit towards its usability.

**Reasons To Reject:**

The evaluation of the proposed tokenization algorithm is not fully conclusive. The proposed method is a combination of two ideas: removal or undertrained tokens (similar to Chizhov et al. 2024) and superword tokens. There are several papers that suggest superword tokenizations too:

https://arxiv.org/pdf/2503.13423
https://ieeexplore.ieee.org/abstract/document/10815724/
https://arxiv.org/abs/2211.11041

It seems natural to provide some form of an ablation study to illustrate if the combination of those two ideas has some particular synergistic effect as well as to compare the proposed solution with the BPE alternatives mentioned in the paper.

Some minor corrections:
"Lian et al. (2024) and Chizhov et al. (2024) proposed strategies to eliminate these low-frequency tokens formed as intermediate steps. We find Chizhov et al. (2024)’s approach particularly compelling due to its direct integration into BPE training."

To the best of my knowledge, both proposed methods integrate the updates into the BPE training direclty, however, Picky BPE has better inference since it takes into consideration the sequence of removals and merges explicitly.

---

> ### Author Response · Authors · 2025-06-02
>
> Overall, we would like to thank you for your detailed review. You clearly spend the time to understand our work, and we appreciate that. Here are more detailed responses to your comments and questions.
>
> > The evaluation of the proposed tokenization algorithm is not fully conclusive. The proposed method is a combination of two ideas: removal or undertrained tokens (similar to Chizhov et al. 2024) and superword tokens. There are several papers that suggest superword tokenizations too:
> <https://arxiv.org/pdf/2503.13423> <https://ieeexplore.ieee.org/abstract/document/10815724/> <https://arxiv.org/abs/2211.11041>
>
> We discuss the relation to the concurrent work by Liu et al (2025) in section 5. Ndomba et all (2024) seems to use WordPiece, starting from a combination of characters and common words.  The WordPiece merges would indeed sometimes create superwords in this case.  However, they appear to not use pre-tokenization, which is a difference from our approach that limits superwords to be a combination of full pretokens. Zhemchuzhina et al (2022) is an interesting analysis of the relation of Zipf’s law and tokenization.
>
> > It seems natural to provide some form of an ablation study to illustrate if the combination of those two ideas has some particular synergistic effect as well as to compare the proposed solution with the BPE alternatives mentioned in the paper.
>
> An ablation study would indeed strengthen the paper. With:
>
> ```text
> BoundlessBPE with PickyBPE and the BOUNDLESS_PATTERN
> ```
>
> as the baseline, we will run:
>
> ```text
> BoundlessBPE with no PickyBPE and the BOUNDLESS_PATTERN
> BoundlessBPE with PickyBPE exactly as the PickyBPE paper and the BOUNDLESS_PATTERN
> BoundlessBPE with PickyPBE and the GPT2_PATTERN
> BoundlessBPE with no PickyPBE and the GPT2_PATTERN
> ```
>
> However, due to the long training time of our current code, we will not have results to share by the end of the response period.
>
> Figure 2 does provide evidence that it is the superwords that provide a larger compression increase than the PickyBPE deletions or the choice of pre-tokenization regex.  The four baseline variants with BPE or PickyBPE and two combinations of regex are essentially on top of each other.  (This is explained by the high fraction of pre-tokens which are represented as single tokens, as in Figure 1).  The y-axis of Figure 2 is the log of the number of merges done at each state of the training. Since each merge decreases the number of tokens by exactly 1, the higher blue curve of Boundless BPE results in more compression.
>
> > Some minor corrections: "Lian et al. (2024) and Chizhov et al. (2024) proposed strategies to eliminate these low-frequency tokens formed as intermediate steps. We find Chizhov et al. (2024)’s approach particularly compelling due to its direct integration into BPE training." To the best of my knowledge, both proposed methods integrate the updates into the BPE training direclty, however, Picky BPE has better inference since it takes into consideration the sequence of removals and merges explicitly.
>
> While not directly stated in our paper, we also use the same change to inference as Picky BPE, in addition to deleting tokens with high IoS during the training process. We will add this information in future versions.
>
> > How would the proposed method affect multilingual models?
>
> The BOUNDLESS_BPE regex offers a small advantage of keeping combining marks \p{M} with other base characters,  This is a flaw in the other 3 regex discussed in Appendix A, although it has been corrected in the GPT4-o regex. The motivations for BoundlessBPE also extend to cases that occur in non-English text, such as non-space delimited scripts and words/graphemes with diacritics/combining marks
>
> > Could you, please, clarify your logic here:
> "We first pretrain 8B models with BPE and SuperBPE tokenizers. We use the OLMO2 7B (OLMo et al., 2024) training configuration, including the model architecture, training hyperparameters, and pretraining corpus, but reduce the total number of training steps to correspond to ∼330B tokens (compared to 4T) for the sake of compute."
> For reference, out of the recent models Qwen 3 600m parameters has 6T tokenizer. Could you explain your reasoning in detail here? Is there any reason why reduced amount of tokens makes your results more conclusive?
>
> The quote you provided is from page 5 of the SuperBPE paper you cite above (Liu et al, 2025), rather than our work.

---

> > ### Comment · Reviewer_sXwb · 2025-06-03
> > **ablation experiments**
> >
> > as far as i understand we have ten more days before the end of the rebuttal period. Do you believe you could share the results (at least some preliminary ones) of the ablation experiments here before the end of this timespan?

---

> > > ### Author Response · Authors · 2025-06-04
> > >
> > > We are running a series of 6 ablation studies on 1GB of data, with a vocabulary size up to 131072.  We are considering 3 variations of PickyBPE.  First, where we break deleted tokens into bytes as in our paper.  Secondly, where it is broken into the pair that created it, as in the original paper, and third with deletions turned off.  For each of these, we are considering our BOUNDLESS_PATTERN, and for a strong baseline the GPT4o regex.
> > >
> > > ```
> > > Tau = 0.9, Picky Deletion to bytes,         regex = Boundless
> > > Tau = 0.9, Picky Deletion to previous pair, regex = Boundless
> > > Tau = 1.1, Picky Deletion off,              regex = Boundless
> > >
> > > Tau = 0.9, Picky Deletion to bytes,         regex = gpt4o
> > > Tau = 0.9, Picky Deletion to previous pair, regex = gpt4o
> > > Tau = 1.1, Picky Deletion off,              regex = gpt4o
> > > ```
> > > Our code saves checkpoint models along the way.  Four of the models are currently at a vocabulary size of around 23k, while the two doing deletions as in the paper have just started running, since we had to implement that option.  While they may not reach 131072 by the Monday end of the discussion period, we will report some preliminary results once we reach a vocab size of 40960.  Hopefully that will be later in the week.
> > >
> > > We expect all of these ablations to have fairly similar performance.  We believe Figure 2 provides evidence that the superwords are having the biggest effect, and that the effects of deletions and regex changes will be much smaller.

---

> > > > ### Author Response · Authors · 2025-06-09
> > > >
> > > > We're in the process of running a set of 6 ablations runs. The larger vocabuary sizes show in our figures are still running, but we have some initial results for a vocabulary size of 32768. We consider 3 types of deletions: We either do PickyBPE deletions as originally described in the PickyBPE paper (where a deletion is split into the two tokens that created it), or we split it into the constituent bytes (as in our paper), or we do no deletions. We consider two pre-tokenization regular expressions: either the Boundless pattern, or the GPT4o pattern.  The six combinations of these cases are shown below:
> > > >
> > > > | Deletion | Regex             | Bytes Per Token   | Renyi Efficiency |
> > > > |----------|-------------------|-------------------|------------------|
> > > > | Original | gpt4o_pattern     | 4.368230          | 0.505951         |
> > > > | Bytes    | gpt4o_pattern     | 4.387356          | 0.506238         |
> > > > | None     | gpt4o_pattern     | 4.355050          | 0.505927         |
> > > > | Original | boundless_pattern | 4.073173          | 0.437042         |
> > > > | Bytes    | boundless_pattern | 4.061793          | 0.437065         |
> > > > | None     | boundless_pattern | 4.061645          | 0.437211         |
> > > >
> > > >  We can see that the type of deletion used has a very minimal impact on either the Bytes Per Token or the Renyi Efficiency.  The pre-tokenization regex has a somewhat larger effect. It is not surprising that the GPT4o pattern has a larger Bytes Per Token.
> > > >
> > > > The [GPT4o pattern](https://github.com/openai/tiktoken/blob/main/tiktoken_ext/openai_public.py#L101-L111) is:
> > > > ```
> > > > pat_str = "|".join([
> > > >     r"""[^\r\n\p{L}\p{N}]?[\p{Lu}\p{Lt}\p{Lm}\p{Lo}\p{M}]*[\p{Ll}\p{Lm}\p{Lo}\p{M}]+(?i:'s|'t|'re|'ve|'m|'ll|'d)?""",
> > > >     r"""[^\r\n\p{L}\p{N}]?[\p{Lu}\p{Lt}\p{Lm}\p{Lo}\p{M}]+[\p{Ll}\p{Lm}\p{Lo}\p{M}]*(?i:'s|'t|'re|'ve|'m|'ll|'d)?""",
> > > >     r"""\p{N}{1,3}""",
> > > >     r""" ?[^\s\p{L}\p{N}]+[\r\n/]*""",
> > > >     r"""\s*[\r\n]+""",
> > > >     r"""\s+(?!\S)""",
> > > >     r"""\s+""",
> > > > ]
> > > > ```
> > > >
> > > > The major difference compared to the Boundless pattern is in the start of the first two branches.  GPT4o allows one other optional character `[^\r\n\p{L}\p{N}]` to combined with a word token.  This results in tokens with an extra byte compared to the Boundless patern, giving a somewhat higher bytes per token. As discussed in Section A.1, this was an intentional decision in the Boundless pattern to keep the Unicode classes more separate.  As discussed in A.1, the evidence about this decision remain mixed.

---

### Official Review · Reviewer_bq5M · 2025-05-13

**Rating:** 6
**Confidence:** 4
**Ethics Flag:** 1

**Summary:**

This submission proposes and analyzes a modification of the popular BPE tokenization algorithm. While the original BPE does not merge across pre-tokenization boundaries (most typically: whitespace), the key idea of the proposed algorithm is to allow "super-merges" of pre-tokens, e.g., merging "_of" and "_the" into "_of_the", thereby creating what the authors call "superwords". After a thorough motivation and description of the proposed algorithm, the algorithm is shown to outperform all common subword tokenization methods in an extensive, but only intrinsic evaluation, which looks at metrics such compression rate and Renyi efficiency.

**Questions To Authors:**

I'm wondering how BoundlessBPE compares to omitting pre-tokenization. In the paper, you say
> Schmidt et al. (2024) demonstrated that entirely omitting pre-tokenization resulted in the poorest downstream performance among 18 evaluated tokenizers.

But looking at Table 1 in Schmidt et al. (2024), it appears that the "no pre-tokenization" setting is only compared for their proposed algorithm, but not for any of the other ones.

Having played around with the --split_by_whitespace=False option offered by SentencePiece, which allows to efficiently merge across whitespace boundaries, I noticed that the resulting vocabulary contains many merges of two, three, or even more pretokens. So I'm wondering how BoundlessBPE compares to naive BPE without pre-tokenization.

Concurrent work (SuperBPE, Liu et al. 2025) has a short paragraph on this, but I'm curious what your take on naive BPE without pre-tokenization is.

**Reasons To Accept:**

1. The idea is simple, well-motivated, and makes sense, and the resulting tokenizer -- at least in the intrinsic evaluation -- performs very well.
2. The paper is written very clearly and also serves as a nice introduction to recent work in this area
3. The intrinsic evaluation is thorough and convincing (in terms of comparing tokenizers on their own)

**Reasons To Reject:**

1. The evaluation is only intrinsic, i.e., there is no evaluation of the proposed algorithm in terms of language modeling or downstream task performance. One of the measures used, Renyi efficiency, may be one of the better metrics for predicting relative downstream task performance, but  as Cognetta et al. (2024), note "Although useful, the predictive power of this metric is not perfect". This submission would be *much* stronger if it would show the benefit of superwords in an extrinsic evaluation, maybe by training -- at the very least -- a NanoGPT-style proof-of-concept. I know that this relates to availability of computing resources, but I still find it difficult to give a higher score without any extrinsic evaluation.


Reference: Marco Cognetta, Vilém Zouhar, Sangwhan Moon, and Naoaki Okazaki. 2024. Two Counterexamples to Tokenization and the Noiseless Channel. In Proceedings of the 2024 Joint International Conference on Computational Linguistics, Language Resources and Evaluation (LREC-COLING 2024), pages 16897–16906, Torino, Italia. ELRA and ICCL.

---

> ### Author Response · Authors · 2025-06-02
>
> # bq5M Response
>
> Overall, we would like to thank you for your detailed review. You clearly spend the time to understand our work, and we appreciate that. Here are more detailed responses to your comments and questions.
>
> > One of the measures used, Renyi efficiency, may be one of the better metrics for predicting relative downstream task performance, but as Cognetta et al. (2024), note "Although useful, the predictive power of this metric is not perfect".  Reference: Marco Cognetta, Vilém Zouhar, Sangwhan Moon, and Naoaki Okazaki. 2024. Two Counterexamples to Tokenization and the Noiseless Channel. In Proceedings of the 2024 Joint International Conference on Computational Linguistics, Language Resources and Evaluation (LREC-COLING 2024), pages 16897–16906, Torino, Italia. ELRA and ICCL.
>
> You’re absolutely right that there is mixed evidence on the effect of Renyi efficiency on downstream performance.  We will add the citations you mention.  We were trying to design a system with a more uniform distribution of tokens, and it does at least provides a quantitative measure of that, but it may not translate to downstream performance.
>
> > I'm wondering how BoundlessBPE compares to omitting pre-tokenization. In the paper, you say Schmidt et al. (2024) demonstrated that entirely omitting pre-tokenization resulted in the poorest downstream performance among 18 evaluated tokenizers. But looking at Table 1 in Schmidt et al. (2024), it appears that the "no pre-tokenization" setting is only compared for their proposed algorithm, but not for any of the other ones. Having played around with the `--split_by_whitespace=False` option offered by SentencePiece, which allows to efficiently merge across whitespace boundaries, I noticed that the resulting vocabulary contains many merges of two, three, or even more pretokens. So I'm wondering how BoundlessBPE compares to naive BPE without pre-tokenization. Concurrent work (SuperBPE, Liu et al. 2025) has a short paragraph on this, but I'm curious what your take on naive BPE without pre-tokenization is.
>
> That is an interesting question.  The main difference is that BoundlessBPE guarantees that it only combines full pretokens into superwords, while this would not be true with BPE without pre-tokenization.
>
> As an experiment, we trained a SentencePiece BPE tokenizer on 100MB of our training data and the `split_by_whitespace=False` parameter.  Due to the slowdown from having no pre-tokenization of the training data it took an hour to train.
>
> Out of a vocabulary size of 131,072, there were 4384 vocabulary words that start with a lowercase letter, and then have a combination of understores and letters (matching `[a-z]+_\w+`, since `\w` includes the underscore).  Here is a random sample of these examples:
>
> ```python
> ['ed▁international',
>  's▁done',
>  'es▁are',
>  'ancy▁and',
>  'ing▁our',
>  's▁un',
>  't▁happen',
>  'ing▁authority',
>  'to▁gg',
>  'span▁class',
>  's▁include',
>  'ed▁an',
>  't▁pass',
>  't▁mean▁to',
>  's▁were▁being',
>  'mathbb▁Q',
>  'up▁of',
>  'ing▁in▁mind',
>  'label▁switch',
>  'ed▁her',
>  'ed▁two',
>  'tered▁with',
>  's▁military',
>  'ise▁the',
>  'ed▁the▁problem',
>  's▁good',
>  'option▁value',
>  's▁on▁line',
>  'ed▁into▁a',
>  'to▁Y',
>  't▁use',
>  'acies▁of',
>  'ing▁to▁an',
>  's▁of▁the▁form',
>  'ing▁with▁an',
>  's▁and▁have',
>  's▁the▁way',
>  'ing▁on▁it',
>  'iss▁albino▁mice',
>  'ization▁of',
>  'ases▁and',
>  's▁of▁these',
>  'ed▁from',
>  'ic▁acids',
>  'ches▁are',
>  're▁now',
>  're▁in▁a',
>  's▁of▁her',
>  'ated▁from',
>  'new▁THREE']
> ```
>
> The first part of the token often seems to match a suffix, although there are a surprising (to us) number of whole second and third words.  In general, this isn’t as many as we were expecting.
>
> Of course, the lack of pre-tokenization can also lead to other tokens combining different categories of characters, such as combinations of digits, spaces, and/or punctuation:
>
> ```python
> ['+00]',
> '▁8-10',
> '▁-▁10.'
> '▁)▁;',
> '_{1}(']
> ```

---

### Official Review · Reviewer_DcHp · 2025-05-19

**Rating:** 6
**Confidence:** 3
**Ethics Flag:** 1

**Summary:**

The paper is generally of good quality. It presents a well-motivated approach to improving BPE tokenization. The empirical evaluation, though focused on intrinsic metrics, is conducted thoroughly across multiple vocabulary sizes and compared against standard baselines. The discussion of pre-tokenization regex patterns in the appendix is detailed and insightful.
The paper is also very well-written and generally clear. The core concept of "supermerges" (the boundless part) is well-explained, with many examples to aid understanding.
The central idea of allowing merges across pre-token boundaries ("supermerges") within a BPE framework is relatively novel. Though similar approaches have been investigates, the lack of additional parameters is a plus.

**Questions To Authors:**

#### PickyBPE

The paper should provide a clearer rationale for its specific variant of PickyBPE, particularly the decision to decompose deleted tokens into bytes rather than their constituent BPE tokens (as in the original PickyBPE). What results or hypotheses led to this choice? The concern that this aggressive decomposition could lead to re-learning identical merges also needs to be addressed more directly.
#### Supermerges restriction pattern

The regex used to restrict which pretokens can form supermerges is very influential but discussed only in a footnote. Its design choices, such as using [a-zA-Z] instead of a more Unicode-aware \p{L}\p{M}* (which is used in the main BOUNDLESS_PATTERN), should be justified. This current pattern appears to limit supermerges to English-like words. The special handling for snake_case variables also seems somewhat arbitrary when other common mixed-text patterns are not explicitly included.

#### Multilinguality

The experiments are conducted on MiniPile, an English-only dataset. This is a notable limitation, as most modern LLMs are multilingual. The paper should clearly state this limitation and ideally discuss how the proposed system (especially supermerges and the capitalization-sensitive BOUNDLESS_PATTERN) might perform or interact with languages that have different orthographic conventions, such as those without spaces. Throughout the paper there are also subtle overstatements with respect to languages such as "Pattern B-1 takes a different language-independent approach".

#### Smaller comments
* Figure 2 is difficult to read due to overlapping lines. Perhaps plotting a relative metric or using a different visualization approach could improve clarity.
- While comprehensive, the WordPiece and Unigram results might be less critical, in light of their lack of recent use, and generally subpar performance by the authors' metrics.
 * "no aggregation can help speed up supermerges" seems an overly strong claim, even if not researching or implementing such novel aggregations is reasonable.
* "it still took 4.7 CPU days to train our 1GB training". lacks context. How does this compare to the training time for the baseline BPE on the same hardware and dataset?
* "The T-2, F-2and P-1 word patterns all have a flaw in the handling of combining marks in the \p{M}class" is correct, but also outdated in light of the GPT-4o update.
* "Any regular expression can be used for pre-tokenization, provided that it matches all of the text in a given Unicode string. Thus, at least one branch of the regex must match each of these Unicode categories." This assumes a 'findall' approach. It might be worth noting that split-based approaches, where non-matched spans become pretokens (as used by e.g. DeepSeek), also exist.
* "The other approach to fix this problem is Unicode normalization" Note that though this is fairly accurate for western languages, not all marks can be removed by Unicode normalization.
* "# B-11, leftover marks, just for bad utf-8" - is inaccurate. These are valid standalone Unicode combining marks. The paper's other discussion referring to them as "isolated marks" or "linguistically ill-formed without a base character" is correct.
* For the superwords pattern, I believe ```\$``` should just be ```$```

**Reasons To Accept:**

The proposed method offers a clear and intuitive way to address the limitations of standard pre-tokenization in BPE. It also appears to improve on some other recent methods by requiring less complex setup. The paper is very well-presented, with clear explanations and illustrative examples that make the concepts accessible. Furthermore, the authors engage thoughtfully with both recent and historical literature in the field.

**Reasons To Reject:**

The paper has one primary weakness that may make it not quite ready for publication at COLM: a significant lack of detailed ablation studies. This makes it difficult to understand the precise impact of each individual modification introduced by the authors.

The authors combines several changes: (1) the core "superword" merges, (2) a new variant of PickyBPE deletions, and (3) two new regular expressions (the main BOUNDLESS_PATTERN for pre-tokenization, and a separate regex that constrains which pretokens can form supermerges). Without ablations, it's hard to tell which of these changes are responsible for the observed improvements in token distribution and compression.

Additionally, the configuration of the "standard BPE" baseline used for comparison is not consistently clear. The paper sometimes implies the use of OpenAI's GPT-2 regex pattern, but its choice over the latest one (GPT-4o) is not clear.

Given that the paper focuses exclusively on tokenizer-level metrics and does not include downstream model performance, a detailed ablation study is even more critical. Such a study would allow readers to better interpret the results and understand the value of each proposed change. An ideal study would incrementally add features to a clearly defined, strong baseline (e.g., standard BPE with a modern regex like GPT-4o's). While this would likely require significant additional work and a resubmission, it would greatly strengthen the paper and clarify the benefits of each component.

---

> ### Author Response · Authors · 2025-06-02
>
> Overall, we would like to thank you for your excellent, detailed review. You clearly spent a great deal of time understanding our work, and we appreciate that. Here are more detailed responses to your comments and questions. Other small suggestions not mentioned in this reply will be incorporated.
>
> > The paper has one primary weakness that may make it not quite ready for publication at COLM: a significant lack of detailed ablation studies.
>
> An ablation study would indeed strengthen the paper. With:
>
> ```text
> BoundlessBPE, our version of PickyBPE, BOUNDLESS_PATTERN
> ```
>
> as the baseline, we will run:
>
> ```text
> BoundlessBPE, no PickyBPE, BOUNDLESS_PATTERN
> BoundlessBPE, PickyBPE exactly as the PickyBPE paper, BOUNDLESS_PATTERN
> BoundlessBPE, our version of PickyPBE, GPT2_PATTERN
> BoundlessBPE, no PickyPBE, GPT2_PATTERN
> ```
>
> However, due to the long training time of our current code, we will not have results to share by the end of the discussion period.  Figure 2 does provide evidence that it is the superwords that provide a larger compression increase than the PickyBPE deletions or the choice of pre-tokenization regex.  The four baseline variants with BPE or PickyBPE and two combinations of regex are essentially on top of each other.  (This is explained by the high fraction of pre-tokens which are represented as single tokens, as in Figure 1).  The y-axis of Figure 2 is the log of the number of merges done at each state of the training, so the higher blue curve of Boundless BPE results in more compression.
>
> > Additionally, the configuration of the "standard BPE" baseline used for comparison is not consistently clear. ...
>
> We will explicitly state the regex used in Figures 4-7, which was the GPT2_PATTERN.  We chose that over GPT-4o as it is the default for the Huggingface BPE implementation, and hence very commonly used.
>
> > PickyBPE The paper should provide a clearer rationale for its specific variant of PickyBPE, particularly the decision to decompose deleted tokens into bytes rather than their constituent BPE tokens (as in the original PickyBPE). What results or hypotheses led to this choice? The concern that this aggressive decomposition could lead to re-learning identical merges also needs to be addressed more directly.
>
> Our hypothesis was that the more aggressive decomposition could allow the deleted token to form more compressed tokens.  It does lead to re-learning some merges, but having duplicate re-learned merges only increases the inference time somewhat. Our ablations will show if this hypothesis was correct.
>
> > *Supermerges* ... Its design choices, such as using [a-zA-Z] instead of a more Unicode-aware \p{L}\p{M}* ..., should be justified.
>
> Using \p{L}\p{M}* rather than [a-zA-Z] is indeed a better choice in this regex, and we’ll update our code to use it.  As our results are from MiniPile, which is primarily English, it should not change our results substantially. We included snake_case handling in this regex - along with the apostrophe - since those are the only two non-letter characters allowed in our “words”.  Including the underscore in this regex allows some snake case variable names to recombine into full variable names, as discussed in section 3.5.
>
> > *Multilinguality* The experiments are conducted on MiniPile, an English-only dataset. This is a notable limitation...
>
> You’re entirely correct that this is a major limitation and it should be explicitly stated as such. We have updated our pattern since this submission to improve handling of non-space delimited scripts by breaking those words into extended grapheme clusters and recombining them as counts suggest into superwords.
>
> > Throughout the paper there are also subtle overstatements with respect to languages such as "Pattern B-1 takes a different language-independent approach".
>
> You’re right that this is worded too strongly. It is an improvement over the hardcoded English contractions in T-1 and F-1.
>
> > Figure 2 is difficult to read due to overlapping lines.
>
> We were intending to show in Figure 2 that the blue Boundless line is above the other 4, which are essentially right on top of each other.  We should mention  that this overlap is caused by the prevalence of pre-tokens as full tokens between these four baselines.
>
> > "it still took 4.7 CPU days to train our 1GB training". lacks context. How does this compare to the training time for the baseline BPE on the same hardware and dataset?
>
> Training Hugginface BPE on the same data, vocab, and machine takes 59s. We are currently working on speeding up our code to allow larger training datasets.

---

> > ### Comment · Reviewer_DcHp · 2025-06-02
> >
> > Thank you for the detailed response.
> > However, I'm not convinced by the justification for using GPT2_PATTERN simply because it's the Huggingface default, which are often just legacy choices or toy examples (e.g. BERT for generative modelling). Comparing against patterns used by high-performing contemporary models (like GPT-4o or DeepSeek) would provide much more meaningful insights.
> > Given your training time constraints and the significant changes to the paper the planned ablations will introduce, it seems deferral to the next cycle would still be most appropriate.

---

> > > ### Author Response · Authors · 2025-06-02
> > >
> > > We could just as easily change the last two ablations to be:
> > >
> > > ```
> > > BoundlessBPE, our version of PickyPBE, GPT4o_PATTERN
> > > BoundlessBPE, no PickyPBE, GPT4o_PATTERN
> > > ```
> > >
> > > if you feel that would be more insightful.

---

> > > > ### Author Response · Authors · 2025-06-04
> > > >
> > > > As an update, we are running a series of 6 ablation studies on 1GB of data, with a vocabulary size up to 131072.  We are considering 3 variations of PickyBPE.  First, where we break deleted tokens into bytes as in our paper.  Secondly, where it is broken into the pair that created it, as in the original paper, and third with deletions turned off.  For each of these, we are considering our BOUNDLESS_PATTERN, and for a strong baseline the GPT4o regex.
> > > >
> > > > ```
> > > > Tau = 0.9, Picky Deletion to bytes,         regex = Boundless
> > > > Tau = 0.9, Picky Deletion to previous pair, regex = Boundless
> > > > Tau = 1.1, Picky Deletion off,              regex = Boundless
> > > >
> > > > Tau = 0.9, Picky Deletion to bytes,         regex = gpt4o
> > > > Tau = 0.9, Picky Deletion to previous pair, regex = gpt4o
> > > > Tau = 1.1, Picky Deletion off,              regex = gpt4o
> > > > ```
> > > > Our code saves checkpoint models along the way.  Four of the models are currently at a vocabulary size of around 23k, while the two doing deletions as in the paper have just started running, since we had to implement that option.  While they may not reach 131072 by the Monday end of the discussion period, we will report some preliminary results once we reach a vocab size of 40960.  Hopefully that will be later in the week.
> > > >
> > > > We expect all of these ablations to have fairly similar performance.  We believe Figure 2 provides evidence that the superwords are having the biggest effect, and that the effects of deletions and regex changes will be much smaller.

---

> > > > > ### Author Response · Authors · 2025-06-04
> > > > >
> > > > > Additionally,  we have updated the regex in the to determine if a token can be involved in a supermerge.  Our previously regex was a byte level regex, which limits its complexity.  We now convert a token to utf-8 strings and use the following string regex
> > > > >
> > > > > ```
> > > > > # - match the entire string
> > > > > # - lookahead ensures there is at least one letter
> > > > > # - otherwise can have spaces, underscores, apostrophes, or curly apostrophes
> > > > > IMPROVED_MERGE_PATTERN = r"^(?=.+\p{L})(?:\p{L}\p{M}*|[ _'\u2019])+$"
> > > > > ```
> > > > > This will be used in the ablation studies.

---

> ### Comment · Reviewer_DcHp · 2025-06-05
>
> Best of luck on the ablations. Small suggestion for a simpler way to write the regex below:
> ```
> IMPROVED_MERGE_PATTERN = r"^(?=.*\p{L})[\p{L}\p{M} _'\u2019]+$"
> ```
> or even
> ```
> IMPROVED_MERGE_PATTERN = r"^\p{L}[\p{L}\p{M} _'\u2019]*$"
> ```

---

> > ### Author Response · Authors · 2025-06-05
> >
> > Your first suggestion is a nice simplification, given that you shouldn't have "leading" \p{M} in practice.  The second suggestion would preclude leading spaces, which are very common in superwords, so that wouldn't work.

---

> > > ### Author Response · Authors · 2025-06-09
> > >
> > > We're in the process of running a set of 6 ablations runs. The larger vocabuary sizes show in our figures are still running, but we have some initial results for a vocabulary size of 32768. We consider 3 types of deletions: We either do PickyBPE deletions as originally described in the PickyBPE paper (where a deletion is split into the two tokens that created it), or we split it into the constituent bytes (as in our paper), or we do no deletions. We consider two pre-tokenization regular expressions: either the Boundless pattern, or the GPT4o pattern.  The six combinations of these cases are shown below:
> > >
> > > | Deletion | Regex             | Bytes Per Token   | Renyi Efficiency |
> > > |----------|-------------------|-------------------|------------------|
> > > | Original | gpt4o_pattern     | 4.368230          | 0.505951         |
> > > | Bytes    | gpt4o_pattern     | 4.387356          | 0.506238         |
> > > | None     | gpt4o_pattern     | 4.355050          | 0.505927         |
> > > | Original | boundless_pattern | 4.073173          | 0.437042         |
> > > | Bytes    | boundless_pattern | 4.061793          | 0.437065         |
> > > | None     | boundless_pattern | 4.061645          | 0.437211         |
> > >
> > >  We can see that the type of deletion used has a very minimal impact on either the Bytes Per Token or the Renyi Efficiency.  The pre-tokenization regex has a somewhat larger effect. It is not surprising that the GPT4o pattern has a larger Bytes Per Token.
> > >
> > > The [GPT4o pattern](https://github.com/openai/tiktoken/blob/main/tiktoken_ext/openai_public.py#L101-L111) is:
> > > ```
> > > pat_str = "|".join([
> > >     r"""[^\r\n\p{L}\p{N}]?[\p{Lu}\p{Lt}\p{Lm}\p{Lo}\p{M}]*[\p{Ll}\p{Lm}\p{Lo}\p{M}]+(?i:'s|'t|'re|'ve|'m|'ll|'d)?""",
> > >     r"""[^\r\n\p{L}\p{N}]?[\p{Lu}\p{Lt}\p{Lm}\p{Lo}\p{M}]+[\p{Ll}\p{Lm}\p{Lo}\p{M}]*(?i:'s|'t|'re|'ve|'m|'ll|'d)?""",
> > >     r"""\p{N}{1,3}""",
> > >     r""" ?[^\s\p{L}\p{N}]+[\r\n/]*""",
> > >     r"""\s*[\r\n]+""",
> > >     r"""\s+(?!\S)""",
> > >     r"""\s+""",
> > > ]
> > > ```
> > >
> > > The major difference compared to the Boundless pattern is in the start of the first two branches.  GPT4o allows one other optional character `[^\r\n\p{L}\p{N}]` to combined with a word token.  This results in tokens with an extra byte compared to the Boundless patern, giving a somewhat higher bytes per token. As discussed in Section A.1, this was an intentional decision in the Boundless pattern to keep the Unicode classes more separate.  As discussed in A.1, the evidence about this decision remain mixed.

---

### Decision · Program_Chairs · 2025-07-08

**Decision:**

Accept

**Comment:**

This tokenization paper proposes to lower the impact of pretokenization, by allowing complete pretokens to be merged (but not smaller units, a major difference of just applying BPE without pretokenization). The authors engaged with the (excellent!) reviews and the discussion reads like a journal submission condensed in timeline. With the additional remarks and experiments the paper will be much stronger

The main drawbacks are:

- lack of dissociation of what modification drove what gains. *This has been addressed during the rebuttal phase*

- no evidence of impact of the final model. This is a general issue with tokenization proposal, and while a pity, it is a lot to ask for such a paper to train even small LLMs (and even then, there will always be the doubt if this extrapolates to larger models)

- lack of multilingual experiments. This seems a major issue, and the main reason for my `Maybe` rating for what is otherwise an excellent paper